# Integrating Clinical Parameters into Thyroid Nodule Malignancy Risk: A Retrospective Evaluation Based on ACR TI-RADS

**DOI:** 10.3390/jcm14155352

**Published:** 2025-07-29

**Authors:** Nikolaos Angelopoulos, Ioannis Androulakis, Dimitrios P. Askitis, Nicolas Valvis, Rodis D. Paparodis, Valentina Petkova, Anastasios Boniakos, Dimitra Zianni, Andreas Rizoulis, Dimitra Bantouna, Juan Carlos Jaume, Sarantis Livadas

**Affiliations:** 1Academic Department of Nuclear Medicine, School of Medicine, AHEPA University Hospital, 546 36 Thessaloniki, Greece; drangelnick@gmail.com; 2Hellenic Endocrine Network, 105 63 Athens, Greece; i-androulakis@hen.gr (I.A.); dimitrios.askitis@gmail.com (D.P.A.); valvisnikos@yahoo.com (N.V.); rodis@paparodis.gr (R.D.P.); valiapetg@gmail.com (V.P.); anbon@otenet.gr (A.B.); dimzianni@hotmail.com (D.Z.); andreasrizoulis@hotmail.com (A.R.); dimitra.bantouna@yahoo.gr (D.B.); 3Loyola University Medical Center, Maywood, IL 60153, USA; juan.jaume@va.gov; 4Edward Hines Jr. VA Hospital, Hines, IL 60141, USA; 5Athens Medical Centre, 117 42 Athens, Greece

**Keywords:** thyroid nodules, malignancy, age, Thyroid Imaging Reporting and Data System

## Abstract

**Background/Objectives**: Thyroid nodules are commonly found through sensitive imaging methods like ultrasonography. While most nodules are benign and asymptomatic, certain characteristics may indicate malignancy, prompting fine needle aspiration biopsy. Factors like age and gender affect cancer risk, complicating ultrasound-based risk systems. We aimed to determine whether the cytological malignancy rate of thyroid nodules could be adjusted for several clinical parameters. **Methods**: Data from patients aged 18 and above with thyroid nodules assessed via fine needle aspiration (FNA) were retrospectively reviewed. Malignancy classification was based on cytopathology and histopathology results. The study examined how various clinical parameters, adjusted for the ACR TI-RADS category, affected thyroid nodule malignancy rates, including age, sex, Body Mass Index (BMI), nodule size, presence of autoimmunity, and thyroxine therapy. Additionally, we analyzed the performance of ACR TI-RADS in predicting malignant cytology across different age subgroups of thyroid nodules. **Results**: The study included 1128 thyroid nodules from 1001 adult patients, with a median age of 48 years and predominantly female (76.68%). Malignancy rates varied across ACR TI-RADS categories, with higher rates associated with larger nodules and younger age groups. Age emerged as a significant predictor of malignancy, with a consistent decrease in the odds ratio for malignant cytology with advancing age across all ACR TI-RADS categories, indicating its potential utility in risk assessment alongside nodule size and sex. **Conclusions**: Raising the size threshold for recommending FNA of TR3-3 nodules and incorporating patients’ age and gender into the evaluation process could enhance the system’s accuracy in assessing thyroid nodules and guiding clinical management decisions.

## 1. Introduction

The prevalence of thyroid nodules varies depending on different diagnostic methods [1]: about 5% of individuals may have thyroid nodules detected during routine physical examinations; however, using more sensitive imaging techniques such as computed tomography (CT) or ultrasonography (US) nodules can be detected in up to 15% to 67% of people, respectively [2]. About 95% of thyroid nodules are asymptomatic and are often discovered incidentally during imaging tests performed for other reasons or during routine physical examinations [3]. While most thyroid nodules are benign, for those with suspicious ultrasound characteristics, a fine needle aspiration (FNA) biopsy is indicated to determine whether the nodule is cancerous.

As age increases, thyroid nodules become more prevalent, but their likelihood of being malignant decreases [4]. It has been proposed that several clinical variables, obtained during routine initial thyroid nodule evaluation, should be considered in assessing malignancy potential: younger age, male sex, and nodule size seem to increase malignancy, and “individualization” on thyroid nodule care is recommended [5].

In clinical practice, the risk of malignancy is assessed using ultrasound-based stratification systems, such as the American College of Radiology Thyroid Imaging Reporting and Data System (ACR TI-RADS) [6] and the European Thyroid Imaging Reporting and Data System (EU-TIRADS) [7]. Of note, in both classification systems, nodule stratification is performed regardless of clinical variables and is based solely on ultrasound characteristics. Demographic factors like age or gender are not taken into consideration. For example, in a subgroup of patients, such as older adults (>70), the specificity of TIRADS was low (28%) and the number of unnecessary biopsies increased [8].

Despite these observations, there is a notable gap in the literature regarding how clinical variables interact with ultrasound-based classifications to affect malignancy risk estimates. In particular, it remains unclear whether incorporating factors such as age could improve the diagnostic precision of TI-RADS categories. Therefore, the primary aim of this study was to assess whether clinical variables—particularly age—modify the cytological malignancy risk of thyroid nodules stratified by the ACR TI-RADS system. We also examined the potential influence of gender, Body Mass Index (BMI), the presence of autoimmune thyroid disease, and thyroxine replacement therapy. We hypothesized that age would significantly influence malignancy rates across specific TI-RADS categories, suggesting a need for age-adapted risk stratification in clinical practice.

## 2. Materials and Methods

### 2.1. Design and Patients’ Characteristics

We performed a retrospective cohort study of patients aged ≥18 years with thyroid nodules evaluated using fine needle aspiration (FNA) at ten endocrine clinics in Greece between January 2023 and January 2024. Cytopathology results from FNA and histopathology results from thyroidectomies were used to classify malignancy. All cases with histologically confirmed malignancy were classified as papillary thyroid carcinoma (PTC); no other histological subtypes were included in the analysis.

Thyroid nodules had been evaluated for composition, margins, echogenicity, shape, presence of echogenic foci, and description of vascularization pattern (color Doppler) in the initial examination by experienced endocrinologists and classified according to the ACR TI-RADS. TIRADS classification was based on the original sonographic report. Cytological findings were reported based on the classification of thyroid nodules according to the criteria of the Bethesda System for Reporting Thyroid Cytopathology into 6 diagnostic categories: (I) nondiagnostic, (II) benign, (III) atypia of undetermined significance (AUS), (IV) follicular neoplasm, (V) suspicious for malignancy (SFM), and (VI) malignant [9].

The exclusion criteria were as follows: Nodules with Bethesda I, III, and IV FNA results, unless cytology was further confirmed by thyroidectomy and operational histopathology reports. This study was approved by the Hellenic Endocrine Network Institutional Review Board (Protocol Number: 2024/0121314). The Institutional Review Board waived the requirement for written informed consent due to the retrospective nature and the minimal risks associated with this study. The study was conducted under the standards of the Declaration of Helsinki.

Data from patients’ records were retrieved, regarding the presence of autoimmune thyroiditis (based on antithyroid antibodies titers) and the potential need for treatment with levothyroxine (LT4) due to hypothyroidism (hypothyroidism was defined as a TSH level greater than 5 mIU/L prior to the initiation of LT4 therapy).

Univariate and multivariate analyses were performed to assess the impact of age, sex, Body Mass Index (BMI), nodule size, presence of autoimmunity, and thyroxine therapy on the malignant cytology rate of thyroid nodules adjusted for the respective ACR TI-RADS category. Secondary analysis included an evaluation of the ACR TI-RADS in the prediction of malignant cytology in different age subgroups of thyroid nodules classified as TR3, TR4, and TR5 (since nodules classified as TR1 and TR2 do not indicate subsequent FNA).

### 2.2. Statistical Analysis

Data are presented as absolute numbers and percentages for categorical variables, and in median and interquartile range (IQR) for variables with skewed distributions. The Mann–Whitney test was employed to compare medians of the non-normally distributed variables. We utilized Pearson’s chi-square test or Yates adjustment for continuity in the study of categorical data.

We used receiver operating characteristic (ROC) analysis to evaluate age as a continuous variable in predicting malignancy, calculating the corresponding sensitivity and specificity. The ideal cutoff age for predicting thyroid nodule malignancy was determined using the Youden index. The Clopper-Pearson technique for binomial distribution was employed to compute confidence bounds for a sample proportion.

For this study, we conducted a univariate analysis to determine the risk variables for malignant cytology, including age, sex, BMI, nodule size, presence of thyroid autoimmune disease, and levothyroxine (LT4) therapy. To determine possible independent risk factors for malignant cytology, a multivariate analytic model was employed. This model utilized logistic regression and considered the ACR TI-RADS categories as a control variable. All thyroid nodules investigated in this study were included in both univariate and multivariate analyses. We further investigated the subgroup of patients where FNA was indicated according to the ACR-TIRADS system.

*p* < 0.05 was used to determine statistical significance and all tests were conducted with a two-tailed approach. The statistical analyses were conducted using MedCalc Statistical Software version 19.2.6 (MedCalc Software Ltd., Ostend, Belgium; https://www.medcalc.org; 2020).

## 3. Results

A total of 1191 nodules from 1005 patients were initially evaluated. Eight patients were excluded due to age (<18 years) and 55 were excluded due to the absence of ACR-TIRADS classification, resulting in 1128 thyroid nodules from 1001 adult patients who were included in the analysis (Figure 1 for nodules with a clinical indication for FNA). Demographic characteristics of the study population are listed in Table 1. The median age was 48 (IQR, 38–59) years and 865 were females (76.68%). The median nodule diameter was 16.1 mm (IQR, 12–23). The percentage of nodules classified as TR1, TR2, TR3, TR4, and TR5 was 0.1% (n = 1), 1.1% (n = 11), 17.7% (n = 182), 63.8% (n = 656), and 27.1% (n = 278), respectively. The corresponding malignancy rates were 0%, 0%, 8.8%, 22.1%, and 74.1%, respectively (Table 1).

Stratification revealed a reduction in the malignant cytology rate with advancing age across the subgroups of 18–39, 40–59, and ≥60 years: 37.8%, 21.8%, and 19.1%, respectively (*X*^2^ (2) = 35.49, *p* < 0.001; Table 2). Details for each Bethesda category are shown in Appendix A.

We also analyzed age as a dichotomous variable and the best cutoff for predicting malignancy was 40.25 years (maximum Youden index: 0.189). The prevalence of malignancy was 45.7% and 26.5% in the <41 (161/352) and ≥41 years (206/776) subgroups, respectively (*p* < 0.001). When adjusted for sex and nodule size, age ≤40.25 years was a risk factor for malignancy compared to age > 40.25 years (odds ratio, OR, 2.26; 95% CI: 1.72–2.97; *p* < 0.001). We conducted a post hoc power analysis to determine whether the sample size was sufficient to detect differences in cytological malignancy rates between age groups in patients where FNA indication exists according to ACR-TIRADS system. With 212 nodules in the younger group (≤40.25 years) and 493 in the older group (>40.25 years), and observed malignancy rates of 48.1% and 27.2% respectively, the calculated statistical power was 100% at a significance level of α = 0.05.

Univariate analysis revealed age, sex, and nodule size as risk factors for malignant cytology. For each year of age, there was a 2.33% reduction in the OR for malignant cytology (95% CI: 1.1%–3.1%; *p* = < 0.001). Male sex was associated with increased risk of malignant cytology (OR 1.47, 95% CI: 1.10–1.95; *p* = 0.009), whereas nodule size was a protective factor with OR of 0.94 (0.93–0.96; *p* < 0.001), while BMI, autoimmune thyroiditis, and the use of thyroxine did not show a significant association with malignancy.

The multivariate logistic regression analysis involving 1128 thyroid nodules, adjusted for ACR TI-RADS categories, sex, and nodule size also showed a 2.5% reduction in the OR for malignant cytology for each year of age (Table 3).

We further analyzed data in the subgroup of nodules (705) where FNA was indicated according to their ACR TI-RADS score and diameter (excluding TI-RADS 1 and 2 categories, Figure 1) to estimate potential predictive factors of malignancy in real-world circumstances.

Backward multiple logistic regression analyses were performed to assess the impact of age, sex, nodule size, BMI, autoimmune thyroiditis, and hypothyroidism on the malignant cytology rate of thyroid nodules adjusted for the respective ACR TI-RADS category (Table 4). Intragroup analysis was performed in each group of ACR TI-RADS (TR3, TR4, and TR5) nodules where FNA was indicated (Table 5, Table 6 and Table 7, respectively).

In ACR TI-RADS 3 category nodules, only age was a strong predictor for malignancy (Table 5). ROC analyses revealed a diameter of >34 mm as the cutoff value with the best sensitivity and specificity (Youden index 0.564, sensitivity 83.3%, and specificity 73.1%, Figure 2).

There was a 4.3% decrease in the odds of malignancy for each year of age increase in patients in the ACR TI-RADS 4 category (Table 6). The odds ratio of 0.374 for hypothyroidism suggested a 62.59% decrease in the odds of the outcome of malignancy in cases without hypothyroidism when the age is the same. For every one-year increase in age, there was a 2.1% decrease in the odds of malignancy for patients in the ACR-TIRADS 5 category (Table 7, Figure 3).

## 4. Discussion

Several studies have investigated the impact of clinical variables on the risk of malignancy in fine needle aspiration biopsy (FNAB) cytology of thyroid nodules. Rago et al. found that male sex, single nodularity, and younger age were independent risk factors for malignancy [10]. Angell et al. identified age as a significant predictor with a cutoff of ≤52 years associated with increased malignancy risk [5]. Belfiore et al. categorized age into decades and observed higher odds ratios of malignancy in patients ≤ 30 years and ≥60 years [11]. Similar to our strategy, previous studies that treated age as a continuous variable reported a 2.2% decrease in the relative risk of malignancy per year [4]. There is a growing body of evidence underlying the role of age in sonographic risk estimate of thyroid nodules. Di Fermo et al. demonstrated that while risk stratification systems are applicable in elderly patients, certain systems may underperform in malignancy detection compared to younger cohorts [12]. Similarly, Grani et al. highlighted that sonographic risk stratification systems, including ACR TI-RADS and EU-TIRADS, can serve effectively as rule-out tools in older adults, though their specificity may be influenced by age-related morphological variations [13]. In our study, the age threshold of 40.25 years was statistically derived using the Youden index to optimize discrimination of cytological malignancy risk. Although this cutoff did not directly match previously reported categorical age brackets, it aligns closely with thresholds identified in the literature—such as the ≤52-year cutoff reported by Angell et al. [5] and the increased malignancy odds at extreme age groups described by Belfiore et al. [11]. These converging findings suggest that age may affect thyroid cancer risk in a non-linear and possibly complex manner. The biological plausibility of such thresholds may relate to cumulative mutational burden, hormonal shifts, or immune-related mechanisms emerging during the fourth and fifth decades of life. Thus, our data further support the hypothesis that age can meaningfully modify malignancy risk even within standardized ultrasound-based classifications.

Walter et al. reported a significant age-related variation in malignant cytology rates suggesting a lower predictive value of high ACR TI-RADS scores in older populations [14]. Complementing this, Grani et al. validated the ACR TI-RADS performance in transition-age individuals, reinforcing its applicability in younger populations while indirectly pointing to the need for age-adapted thresholds or interpretations in older adults. Collectively, these findings support the integration of age as a modifying factor in the interpretation of ultrasound risk stratification for thyroid nodules [15].

These aforementioned findings, along with our own, highlight the importance of considering age as a key factor in the evaluation and management of thyroid nodules—particularly in cases with suspicious ultrasound features (TR4 and TR5).

Our study also aimed to evaluate the influence of other clinical variables on the malignant cytology rates within the ACR TI-RADS categories, such as BMI, use of thyroxine, and history of autoimmune thyroiditis. Some research suggested that higher BMI may be associated with an increased risk of thyroid cancer, particularly in women [16]. It is hypothesized that this could be due to various factors such as hormonal changes, obesity-related inflammation, and insulin resistance, which are more prevalent in individuals with higher BMI [17]. However, as is the case in our study, higher BMI was found not to be associated with more aggressive tumor features or a greater likelihood of recurrence or persistence over the analyzed period [17].

It is generally believed that higher TSH concentrations are a risk factor for the growth of thyroid nodules and cancer in adults [18], especially in the absence of autoimmunity [19]. Under this aspect, suboptimal thyroxine supplementation may interfere with the prevalence of thyroid malignancy although such a hypothesis has not been demonstrated clinically. Of note, in our cohort, thyroxine use was recommended as a replacement therapy and not as a suppressive therapy for nodular disease. However, TSH levels were not consistently available and were therefore not included in the final analysis. In the subgroup of patients with TR4, the prevalence of malignancy was lower in those with hypothyroidism. Several studies have reported an increased risk of thyroid cancer in patients with primary hypothyroidism [20,21] while others have not [22]. Whether chronic lymphocytic thyroiditis (CLT) is an important risk factor for differentiated thyroid cancer (DTC) remains a topic of debate. Patients with CLT had an increased risk of incidental PTC in studies that investigated surgical specimens [23,24]. Nevertheless, TPO-Ab and Tg-Ab may play a different role in thyroid malignancy. Another study reported that TgAb positivity was significantly associated with an increased risk of papillary thyroid carcinoma (PTC) in females with an OR of 3.532 (95% CI: 1.219–10.233), whereas TPOAb did not show a significant association [25]. In our study, the presence of at least one of the two antithyroid antibodies was considered indicative of autoimmune thyroiditis.

Regarding age-related variation of DTC prevalence, our study has three major strengths compared to the previous report of Walter et al. [14]. First, ACR TI-RADS stratification was performed by each investigator (trained endocrinologist) during the initial comprehensive “real-time” ultrasound examination and was not based on retrospectively examined static images. It is well known that risk stratification systems (RSS) are subject to interobserver variability, which may influence how nodules are categorized [26,27] with agreement ranging from poor to excellent. However, recent reports illustrated an excellent rating agreement between experienced observers [28], such as the well-trained endocrinologists participating in our study. Second, except for ACR TI-RADS categories, we further analyzed the specific predictive factors of malignancy in each ACR TI-RADS group separately. Third, in our cohort, the proportion of undefined cytology was eliminated since we included only patients with Bethesda III and IV in whom a definitive histopathology diagnosis was available after thyroidectomy. However, this criterion inherently favors the selection of surgically managed cases, which are more likely to represent higher-risk nodules. As a result, the malignancy rates observed in these categories may be overestimated and not entirely reflective of the broader population of Bethesda III/IV nodules encountered in routine clinical practice. This selection bias limits the generalizability of our findings and highlights the need for prospective studies that include conservatively managed nodules within these cytological categories.

Although histopathology after thyroid surgery remains the gold standard of the diagnosis of malignancy, FNA is still the first step to evaluate the malignancy potential with significant diagnostic accuracy with sensitivity and specificity ranging between 65 and 98% and 72 and 100%, respectively [29]. As mentioned, it should be noted that our cohort has a limited number of undefined cytology, since in such cases, either FNA was repeated or surgery was performed, diminishing the effect of undetermined samples on the interpretation of our results. Compared to previous studies, the prevalence of suspicious FNA results was higher in our patients [14]. The main difference was observed in the ACR TI-RADS 3 category (15.2% compared to <2.5%). This alteration may be attributed to our focused assessment of the echogenicity of the nodules. Indeed, the assignment of hypoechogenicity ranged from 19 to 69% in a multi-institutional registry [30]. Assignment of hypoechoic or very hypoechoic echogenicity adds a further 2 to 3 points, moving a TR2 nodule that does not warrant FNA to TR3 or even TR4 category. In this subgroup of patients, nodule size heralds further investigation with FNA. We observed a threshold of 34 mm in nodule size, associated with a sensitivity of 83.3% and specificity of 73.1% for thyroid malignancy. This finding suggests that the currently recommended 25 mm cutoff for FNA indication in TR3 nodules may merit further evaluation in future prospective studies.

The ACR TI-RADS White Paper predicts cancer risks of <2%, <5%, 5–20%, and >20% in TR1–2, TR3, TR4, and TR5 thyroid nodules, respectively [6]. Here, we observed higher malignancy rates comparable to those reported in the ACR TI-RADS guidelines: <1.0%, 15%, 31%, and 86% for TR1-2, TR3, TR4, and TR5, respectively. This should be attributed to our selection criteria since over 60% of the enrolled patients in our study fulfilled evidence-based indications for FNA, based on ACR-TIRADS criteria, while other studies might have involved more specimens from unnecessarily performed FNA biopsies.

In the present study, 423 nodules (37.5%) underwent FNA without a clear indication according to the ACR TI-RADS published guidelines; 79 (7%) of them were small nodules (<10 mm) but with suspicious ultrasound characteristics. Clinicians performed FNA for another 129 (30.5%) nodules with diameters >10 mm without following the ACR TI-RADS published guidelines. This discrepancy is in line with other reports [31] showing that using an RSS reduces unneeded biopsies (57 from 100) compared to using published ACR TI-RADS guidelines (80/100). Further, in a prospective analysis of 502 nodules, Grani et al. found that the “right” TI-RADS avoided unnecessary biopsies in more nodules than other systems [32].

Our study has some limitations that should be discussed. The histotype of thyroid cancer can influence the pre-operative neck US results, also when several RSS were considered, including ACR-TIRADS [33]. In the present study, only papillary thyroid carcinoma (PTC) cases were assessed. This should be stated as a limitation, as it may affect the generalizability of the findings. Another limitation of our study is that no comparison was made among cytologists’ diagnoses across different centers and no interobserver agreement coefficient was calculated. This may have affected the consistency and reliability of the cytological evaluations. In our study, semantic sonographic features were evaluated based on standard descriptive terms and standardized sonographic lexicon was not applied for the definition of semantic ultrasound features, which could have affected the consistency and reproducibility of the results [34,35]. Additionally, no formal assessment of interobserver variability was performed for the ACR TI-RADS classification of nodules, despite the well-known variability in sonographic interpretation among observers. To mitigate this, all ultrasound evaluations were conducted by board-certified endocrinologists with experience in thyroid imaging, following the official ACR TI-RADS guidelines. Internal training sessions were held at the beginning of the study to align classification practices across centers. Nevertheless, the absence of objective inter-rater agreement analysis remains a limitation and should be addressed in future prospective research. Lastly, as previously reported, in several nodules, fine needle aspiration cytology (FNAC) was performed without strict adherence to the indications of the risk stratification system (RSS) used. Although we did not conduct a separate analysis excluding these cases, future studies should consider doing so to more accurately reflect the real-world applicability and diagnostic performance of the system. For example, the malignancy rate observed for TR3 nodules in our study (~15%) is substantially higher than the <5% rate reported in the ACR TI-RADS guidelines. This likely reflects a referral and diagnostic bias, as our cohort included only nodules that underwent FNA or surgery, potentially skewing the risk estimate toward higher values.

## 5. Conclusions

Modifications to the current ACR TI-RADS may be warranted as new data become available. Potential modification could involve adjusting point assignments for certain features [36] and nodule location [37]. Given the typically slow-growing nature of thyroid cancer and the success of active surveillance strategies, our findings raise the hypothesis that increasing the size threshold for recommending FNA of TR3 nodules—particularly in certain subgroups—may help reduce unnecessary procedures. Moreover, incorporating clinical variables such as patient age and gender into risk assessment could potentially enhance the performance of the ACR TI-RADS system, as both our data and previous studies [14] have suggested higher malignancy risk in younger individuals and males. However, these observations should be interpreted with caution and viewed as hypothesis-generating. Prospective, multicenter studies are needed to validate whether such modifications can improve risk stratification and clinical decision-making (Figure 4).

## Figures and Tables

**Figure 1 jcm-14-05352-f001:**
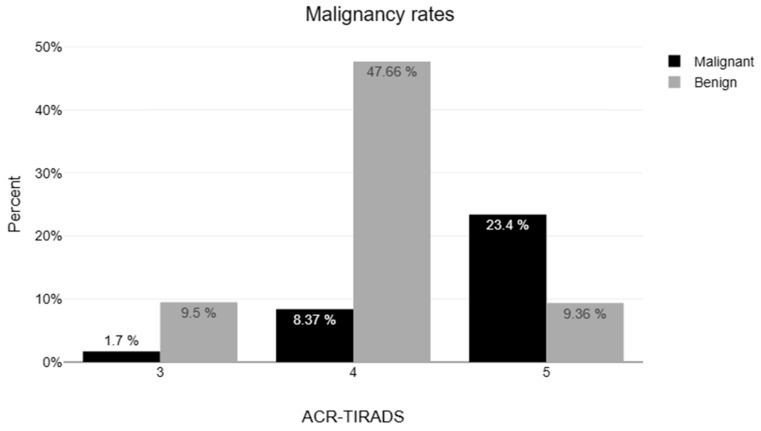
Stratification of the 705 thyroid nodules with an indication for FNA in ACR-TIRADS categories. ACR TI-RADS: American College of Radiology Thyroid Imaging Reporting and Data System. FNA: fine needle aspiration. Percentile refers to the total study population.

**Figure 2 jcm-14-05352-f002:**
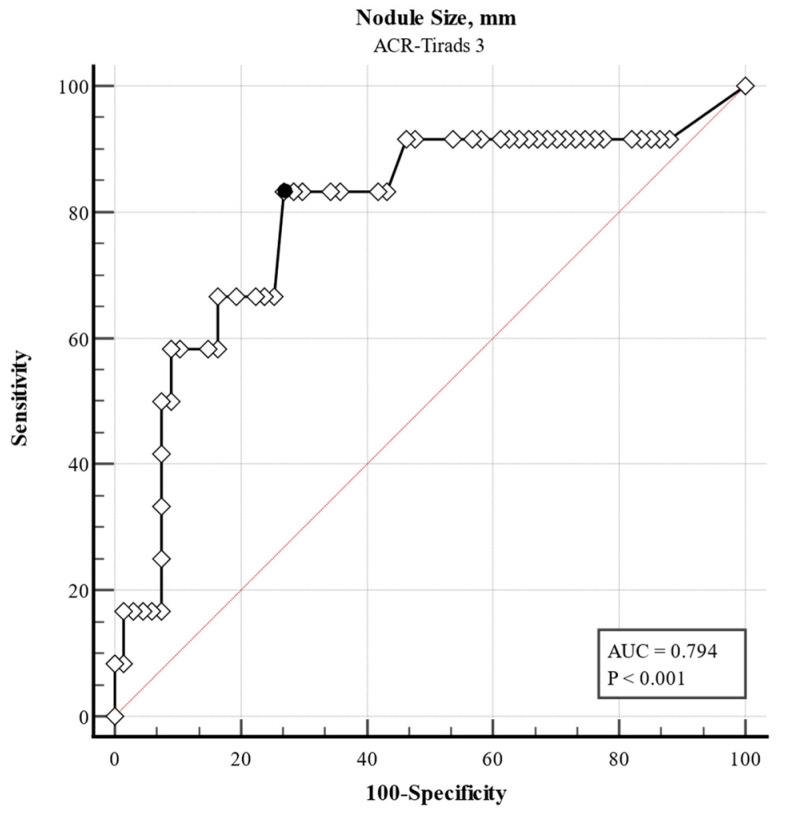
Receiver operating characteristic (ROC) curve illustrating the diagnostic performance of nodule size in predicting cytological malignancy among thyroid nodules classified as ACR TI-RADS category 3 (TR3). The analysis identified an optimal size cutoff of 34 mm, yielding a sensitivity of 83.3% and specificity of 73.1% (area under the curve (AUC) = 0.794). The curve is based on data from 182 nodules with an indication for fine needle aspiration (FNA). Prevalence of malignancy 15.2% (reported by the guidelines: <5%). Youden index 0.564. Abbreviations: ROC, receiver operating characteristic; AUC, area under the curve; TI-RADS, Thyroid Imaging Reporting and Data System; FNA, fine needle aspiration. Black dot = Youden Index.

**Figure 3 jcm-14-05352-f003:**
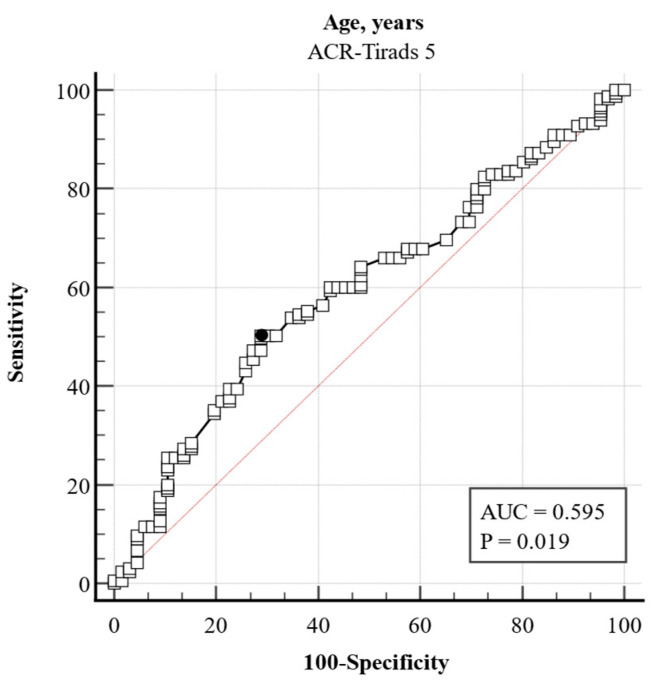
Receiver operating characteristic (ROC) curve assessing the diagnostic value of patient age in predicting cytological malignancy among thyroid nodules classified as ACR TI-RADS category 5 (TR5). The optimal age cutoff, determined by the Youden index (0.215), was ≤43 years. This threshold yielded a sensitivity of 50.3% and specificity of 71.2% for malignancy detection (AUC and 95% CI not shown). Data are derived from a retrospective analysis of 278 nodules evaluated for fine needle aspiration (FNA). Abbreviations: ROC, receiver operating characteristic; TI-RADS, Thyroid Imaging Reporting and Data System; FNA, fine needle aspiration. Black dot = Youden Index.

**Figure 4 jcm-14-05352-f004:**
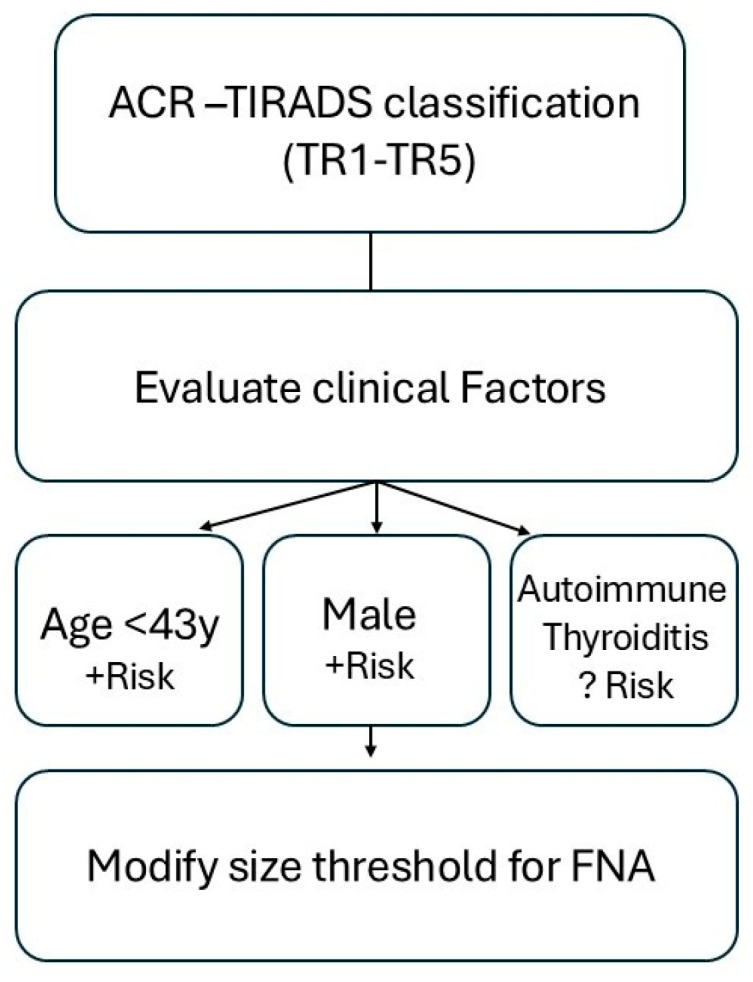
Proposed framework for integrating clinical variables into ACR TI-RADS-based malignancy risk estimation.

**Table 1 jcm-14-05352-t001:** Characteristics of the study population when categorized based on the ACR TI-RADS risk stratification.

**Age, Years**	48 (38–59)	
**Male sex, n (%)**	233 (23.32)	
**Nodule size (mm)**	16.1 (12–23)	
**BMI (kg/m^2^)**	27.2 (23.9–32)	
**Autoimmune thyroiditis, n (%)**	269 (23.85)	
**Hypothyroidism, n (%)**	367 (32.54)	
**Indication for FNA, n (%)**	705 (62.5)	
**ACR TI-RADS categories, n (%)**		**Malignant histopathology, n (%)**
**1**	1 (0.9)	0
**2**	11 (1.1)	0
**3**	182 (17.7)	16 (8.8)
**4**	656 (63.8)	145 (22.1)
**5**	278 (27.1)	206 (74.1)
**Bethesda categories, n (%)**		
**I**	11 (0.9)	1 (9)
**II**	531 (47.1)	9 (1.7)
**III**	262 (23.2)	72 (27.5)
**IV**	33 (2.9)	5 (15.2)
**V**	145 (12.8)	136 (93.8)
**VI**	146 (12.9)	144 (98.6)

**Table 2 jcm-14-05352-t002:** Age stratification, FNA results, and thyroid malignancy.

Bethesda Category	20–39 y (n = 328)	40–59 y (n = 528)	≥60 y (n = 272)	*p*-Value
Benign (I–II)	110 (33.5%)	272 (51.5%)	160 (58.9%)	<0.001
Indeterminate (III–IV)	94 (28.7%)	141 (26.7%)	60 (22.1%)	0.116
Malignant (V–VI)	124 (37.8%)	115 (21.8%)	52 (19.1%)	<0.001
**Confirmed malignancy** *	150 (45.7%)	153 (28.9%)	64 (23.5%)	<0.001

Abbreviations: y: age in years, FNA: fine needle aspiration, V, VI: includes both Bethesda V (suspicious for malignancy) and Bethesda VI (malignant) categories, *: malignancy based on the histopathological results.

**Table 3 jcm-14-05352-t003:** Risk factors for malignancy, adjusted for the ACR-TIRADS categories.

	Malignancy			Univariate Analysis		Multivariate Analysis	
	Yes (n = 367)	No (n = 761)	Coefficient B	Odds Ratio (95% CI)	*p*	Odds Ratio (95% CI)	*p*
**Age, years** **(median, IQR)**	44 (35–55)	50 (40–60)	−0.02	0.98 (0.97–0.99)	<0.001	0.98 (0.97–0.99)	<0.001
**Nodule size, mm**	13 (10.65–19.2)	18 (13–24)	−0.06	0.94 (0.93–0.96)	<0.001	0.96 (0.95–0.99)	0.01
**Male sex, N ***	92/324 (28.4)	141/677 (20.8)	0.38	1.47 (1.10–1.95)	0.009	1.42 (0.99–2.01)	0.051
**AT, n (%)**	92 (25.1)	177 (23.3)			0.504		
**Hypothyroidism, n (%)**	112 (30.5)	255 (33.5)			0.315		
**BMI, kg/m^2^**	27.00 (23.6–32.42)	27.34 (23.92–31.96)			0.814		

Univariate and multivariate analyses were conducted to assess the influence of age, sex, multinodularity, and nodule size as risk factors for malignant cytology in 1128 thyroid nodules from 1001 patients. These analyses were adjusted for the ACR TI-RADS categories. Data are expressed as median and interquartile range (IQR) and percentiles. Abbreviations: ACR TI-RADS: American College of Radiology Thyroid Imaging Reporting and Data System. AT: autoimmune thyroiditis, BMI: Body Mass Index, n: number of nodules, N *: refers to the number of patients.

**Table 4 jcm-14-05352-t004:** Predictors of malignancy in 705 nodules with indication for FNA according to ACR-TIRADS system.

	Coefficient B	Std. Error	z	OR	95% CI	*p*
Age, years	−0.03	0.01	4.19	0.97	0.96–0.98	<0.001
Hypothyroidism	−0.52	0.24	2.18	0.6	0.37–0.95	0.029
Male sex	0.14	0.24	0.58	1.15	0.72–1.83	0.564
BMI, kg/m^2^	0.02	0.02	0.92	1.02	0.98–1.05	0.359
AT	−0.19	0.26	0.72	0.83	0.5–1.38	0.472
Nodule size, mm	0.01	0.01	0.84	1.01	0.98–1.04	0.401

Abbreviations: ACR TI-RADS: American College of Radiology Thyroid Imaging Reporting and Data System. AT: autoimmune thyroiditis, BMI: Body Mass Index, SE: standard error, OR: odds ratio, CI: coefficient interval.

**Table 5 jcm-14-05352-t005:** Backward logistic regression analysis of predictive factors for malignancy in ACR-TIRADS 3 category nodules.

Variable	Coefficient	Std. Error	Wald	OR	95% CI	*p*
Nodule size, mm	0.16250	0.049951	10.5829	1.176	1.066–1.297	0.001

Significant overall model fit, *p* < 0.001. Variables not included in the model: BMI, Hashimoto, age, hypothyroidism, and gender.

**Table 6 jcm-14-05352-t006:** Backward logistic regression analysis of predictive factors for malignancy in ACR-TIRADS 4 category nodules.

Variable	Coefficient	Std. Error	Wald	OR	95% CI	*p*
Age	−0.043998	0.011991	13.4629	0.9570	0.934–0.979	0.001
Hypothyroidism	−0.98317	0.39491	6.1981	0.3741	0.172–0.811	0.012

Significant overall model fit. *p* < 0.001. Variables not included in the model: BMI, Hashimoto, nodule size, sex. Prevalence of malignancy is 14.9% (reported by the guidelines: 5–20%).

**Table 7 jcm-14-05352-t007:** Backward logistic regression analysis of predictive factors for malignancy in ACR-TIRADS 5 category nodules.

Variable	Coefficient	Std. Error	Wald	Odds Ratio	95% CI	*p*
Age	−0.020841	0.010382	4.0295	0.9794	0.9596–0.9995	0.044

Significant overall model fit, *p* = 0.043. Variables not included in the model: BMI, Hashimoto, hypothyroidism, sex, nodule size. Disease prevalence is 71.4% (reported by the guidelines: >20%).

## Data Availability

Data are available from the corresponding author upon reasonable request.

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
