# Peer review of "Integrating Clinical Parameters into Thyroid Nodule Malignancy Risk: A Retrospective Evaluation Based on ACR TI-RADS"

_jcm, 2025, doi:10.3390/jcm14155352_

Round 1

Reviewer 1 Report

Comments and Suggestions for Authors

This retrospective multicenter study evaluates whether age, sex, nodule size, BMI, autoimmunity, and thyroxine therapy impact the malignancy risk of thyroid nodules, with a specific focus on how these clinical variables interact with the ACR TI-RADS risk classification. The study includes over 1100 nodules and offers a substantial dataset to investigate predictive associations beyond sonographic features alone.

The study leverages data from ten endocrine clinics, enhancing the external validity and clinical relevance of its findings. The analysis of age, sex, and nodule size alongside ACR TI-RADS addresses a recognized gap in risk stratification systems, which often exclude non-imaging features.

  1. While nodules classified as Bethesda III and IV were included only if histopathology was available, this may introduce selection bias, as these cases likely represent surgically managed, higher-risk nodules. The authors acknowledge this, but it limits the generalizability of their risk estimates.
  2. Although the study involved expert performing real-time TI-RADS classification, interobserver variability is a known concern in ultrasonographic assessments. No inter-rater agreement (e.g., kappa statistic) was provided.
  3. The reported malignancy rate for TR3 nodules (~15%) exceeds expectations based on ACR guidance (<5%). This may reflect referral or diagnostic bias, and limits the applicability of the proposed size threshold modification for TR3 nodules.
  4.  A visual representation of how clinical variables could be integrated into ACR TI-RADS risk thresholds would enhance usability for practicing clinicians.

Reviewer 2 Report

Comments and Suggestions for Authors

Dear Authors,

Thank you for the opportunity to review your manuscript titled "Integrating Clinical Parameters into Thyroid Nodule Malignancy Risk: A Retrospective Evaluation Based on ACR TI-RADS." The topic is clinically relevant and timely, especially as clinicians continue to refine the use of sonographic classification systems in thyroid nodule risk stratification. Your effort to integrate demographic and clinical variables into the ACR TI-RADS framework reflects a valuable direction in thyroid imaging research. That said, I offer several major and minor comments below to help strengthen the scientific rigor, clarity, and interpretability of the study.

Major Comments

  1. Novelty and Framing
    While the integration of clinical variables into thyroid nodule risk stratification has been previously discussed in the literature, your focus on age as a modifying factor within the TI-RADS system offers modest novelty. However, the introduction would benefit from a clearer delineation of the existing gap this study addresses. Currently, the rationale for the research is somewhat diffuse. Please consider expanding the final paragraph of the introduction to more precisely define your study’s aims and hypotheses.

  2. Justification for Off-Guideline FNAs
    A notable proportion of nodules (37.5%) underwent fine-needle aspiration despite lacking indication under ACR TI-RADS guidelines. While this reflects real-world practice, the manuscript does not sufficiently justify this deviation. Were these decisions based on patient preference, referring physician discretion, or clinical suspicion beyond ultrasound appearance? This must be explained more clearly, both for ethical transparency and to help readers interpret your findings.

  3. Ultrasound and Cytology Interpretation Standards
    The ACR TI-RADS classifications were performed by multiple endocrinologists at various centers, yet there is no mention of interobserver variability assessment. Given the well-documented variability in sonographic scoring, this is a significant omission. At minimum, please discuss this limitation in greater detail. If any measures were taken to ensure consistency (e.g., training sessions, consensus reading), these should be stated explicitly.

  4. Statistical Rationale and Power
    While the statistical analyses are comprehensive, the manuscript would benefit from the inclusion of a power calculation to support the sample size adequacy. Moreover, dichotomization of age at 40.25 years—though statistically derived via the Youden index—should be clinically justified. Why is this threshold meaningful in clinical practice, and is it consistent with existing literature?

  5. Overuse of Self-Citations
    Several key claims regarding the relationship between autoimmune thyroid disease and malignancy risk are supported primarily by the authors' previous publications. While these references may be relevant, the appearance of citation bias could be mitigated by incorporating more independent sources that either validate or contrast with your findings.

  6. Conclusion Overreach
    The suggestion that ACR TI-RADS thresholds should be modified based on your findings may be premature. While your data are compelling, the study remains retrospective and single-cohort in design. The recommendation to raise the FNA size threshold in TR3 nodules, for example, should be framed more cautiously—as a hypothesis-generating observation that warrants prospective validation.

Minor Comments

  • Please revise several awkward or unclear sentences throughout the manuscript. For instance, phrases such as “raising questions for a potential modification” could be better expressed with clearer intent.

  • Some tables (e.g., Table 2) are overly dense and may benefit from reformatting or partial migration to supplementary materials.

  • Ensure that all figure legends are fully self-contained and provide enough detail to interpret the images independently of the main text.

  • Consider clarifying whether only papillary thyroid carcinoma (PTC) cases were included, and if so, state this earlier in the Methods.

Comments on the Quality of English Language

The manuscript is generally understandable; however, the quality of English language and grammar requires improvement. Several sections contain awkward phrasing, inconsistent verb tenses, and minor syntactic errors that affect the clarity and fluency of the narrative. A thorough revision by a professional English-language editor or a native speaker with experience in academic medical writing is strongly recommended to ensure the manuscript meets the linguistic standards of a high-impact journal.

Round 2

Reviewer 2 Report

Comments and Suggestions for Authors

Thank you for your thoughtful and comprehensive revisions. The revised manuscript demonstrates clear improvement in structure, scientific transparency, and clinical framing. The inclusion of more independent references, a clarified study aim, and enhanced methodological detail strengthens the manuscript significantly. Below are specific comments intended to help further refine your work:

1. Introduction and Framing

You have made commendable improvements to the introduction, particularly in defining the study rationale and hypothesis related to the role of age in modifying TI-RADS performance. However, the clinical relevance of the statistically derived age cutoff (40.25 years) remains somewhat abstract. Consider reinforcing this with more robust clinical context—e.g., referencing guideline thresholds, age-related thyroid cancer biology, or practice implications for screening and follow-up.

2. Methods

The overall methodology is sound, and the additional details added in this revision (e.g., post hoc power analysis, inclusion of autoimmune markers, and clarified cytological/histological correlation) improve transparency. Nonetheless:

  • The justification for off-guideline FNAs—while partially addressed—would benefit from more concrete explanation. Were specific clinical factors, institutional protocols, or patient requests involved? This clarity will help readers interpret your outcome data more confidently.

  • As discussed in your reply, the absence of interobserver agreement analysis (e.g., kappa statistics) remains a limitation. It is appreciated that you acknowledged this; however, even a limited assessment (e.g., subset review, consensus scoring) would strengthen confidence in the ultrasound scoring consistency across centers.

3. Results and Interpretation

The results are well presented, with ROC curves and multivariate analyses clearly reported. The migration of more granular cytology data to the supplementary material (Table S1) is a practical and effective revision. The discussion of Bethesda categories and risk stratification by age group is particularly helpful.

Still, one potential area for enrichment would be more explicit presentation of clinical implications. For instance, in TR3 nodules among younger patients, how might this inform decision-making in everyday practice? Consider briefly integrating a clinical decision-support angle or visual aid, even if only speculative or hypothesis-generating.

4. Discussion and Conclusion

The revised discussion is more cautious and better reflects the limitations of your retrospective design. Your acknowledgment of overreach in prior conclusions is appreciated. That said:

  • Phrases suggesting guideline modification (e.g., altering FNA thresholds) should remain clearly framed as exploratory findings that warrant validation. Using terms like “suggests a potential refinement” or “hypothesis-generating finding” throughout would ensure your conclusions remain proportionate to the study design.

  • The discussion of autoimmune thyroiditis and its association with malignancy is improved with broader citation support. However, briefly discussing potential biological mechanisms (e.g., chronic inflammation or TSH stimulation) would add depth and educational value.

5. Language and Clarity

The overall readability has improved, but there are still some syntactic and grammatical issues, particularly in the Methods and Discussion sections (e.g., tense agreement, awkward phrasing). A final round of professional English editing is recommended to ensure fluency and precision throughout the manuscript.

6. Figures and Tables

Figures 2 and 3 are informative and now better explained. Figure legends have been enhanced to be self-contained, which is appreciated. Table 2’s streamlined structure improves clarity. You may still consider using bolding or color-coding to draw attention to statistically significant comparisons or clinically relevant cutoffs, if permitted by the journal.

Summary

This study adds clinically relevant insight into how age modifies the malignancy risk associated with thyroid nodules classified under ACR TI-RADS. With a few final refinements in clarity, tone, and framing, the manuscript is likely to make a useful contribution to the literature on thyroid cancer risk stratification. I appreciate the rigor and responsiveness with which you have addressed prior feedback.

Comments on the Quality of English Language

The manuscript has undergone meaningful revision and is generally understandable; however, the quality of English still requires improvement to ensure clarity, precision, and fluency. Several sections—particularly in the Methods and Discussion—contain awkward phrasing, inconsistent verb tenses, and complex sentence constructions that may hinder reader comprehension.

A thorough language review by a native English speaker with experience in academic medical writing or a professional editing service is strongly recommended. This will help align the manuscript with the linguistic standards expected in high-impact scientific journals.
